# The Impact of the Tumor Microenvironment on Macrophage Polarization in Cancer Metastatic Progression

**DOI:** 10.3390/ijms22126560

**Published:** 2021-06-18

**Authors:** Huogang Wang, Mingo M. H. Yung, Hextan Y. S. Ngan, Karen K. L. Chan, David W. Chan

**Affiliations:** Department of Obstetrics & Gynaecology, LKS Faculty of Medicine, University of Hong Kong, Hong Kong, China; wanghuog@connect.hku.hk (H.W.); h1094157@connect.hku.hk (M.M.H.Y.); hysngan@hku.hk (H.Y.S.N.)

**Keywords:** tumor microenvironment, macrophage polarization, peritoneal metastasis, tumor-associated macrophages, premetastatic niche

## Abstract

Rather than primary solid tumors, metastasis is one of the hallmarks of most cancer deaths. Metastasis is a multistage event in which cancer cells escape from the primary tumor survive in the circulation and disseminate to distant sites. According to Stephen Paget’s “Seed and Soil” hypothesis, metastatic capacity is determined not only by the internal oncogenic driving force but also by the external environment of tumor cells. Throughout the body, macrophages are required for maintaining tissue homeostasis, even in the tumor milieu. To fulfill these multiple functions, macrophages are polarized from the inflammation status (M1-like) to anti-inflammation status (M2-like) to maintain the balance between inflammation and regeneration. However, tumor cell-enforced tumor-associated macrophages (TAMs) (a high M2/M1 ratio status) are associated with poor prognosis for most solid tumors, such as ovarian cancer. In fact, clinical evidence has verified that TAMs, representing up to 50% of the tumor mass, exert both protumor and immunosuppressive effects in promoting tumor metastasis through secretion of interleukin 10 (IL10), transforming growth factor β (TGFβ), and VEGF, expression of PD-1 and consumption of arginine to inhibit T cell anti-tumor function. However, the underlying molecular mechanisms by which the tumor microenvironment favors reprogramming of macrophages to TAMs to establish a premetastatic niche remain controversial. In this review, we examine the latest investigations of TAMs during tumor development, the microenvironmental factors involved in macrophage polarization, and the mechanisms of TAM-mediated tumor metastasis. We hope to dissect the critical roles of TAMs in tumor metastasis, and the potential applications of TAM-targeted therapeutic strategies in cancer treatment are discussed.

## 1. Introduction

Despite advances in cancer treatment, cancer metastasis remains the principal cause of cancer-related deaths. Metastasis is a process in which malignant cells spread beyond the primary tumor to a distal region to cause the development of new secondary tumors or throughout the whole body in advanced cancers. In fact, metastasis develops through multistage procedures, including the detachment of cancer cells from the primary tumor origins, intravasation of tumor cells into the circulatory and lymphatic systems, immune evasion, extravasation to distant capillary beds, and invasion and proliferation in distant organs in the majority of solid tumors [1]. Most solid tumors prefer hematogenous metastasis. For example, colon and breast cancer cells go through multiple steps of intra- and extravasation before establishing tumor dissemination to other organs (e.g., bone, liver, brain). However, metastasis is relatively more straightforward for some cancers, particularly ovarian carcinomas. Metastatic ovarian cancer cells usually diffuse in the abdominal cavity through typical and interrelated processes, including the following events: (a) cancer cell detachment, survival, and resistance to anoikis; (b) evasion of immunological surveillance; (c) epithelial-mesenchymal transition (EMT); (d) spheroid formation; (e) ascites formation; and (f) micro- and macro-implantation [2,3].

Metastasis requires malignant cells to detach from the primary tumor initially [4] and is involved in the development of a premetastatic niche. For instance, ovarian cancer cells depart from the primary ovarian lesion, and they form ascites spheroids in the abdominal cavity and remodel stromal cells and the extracellular matrix to become tumor-supportive and tumor-receptive tissue microenvironments [5]. Recent evidence additionally supports the notion that metastatic cancer cells can alert premetastatic site components, such as primary residential immune cells, to exert a protumor effect that facilitates metastasis [6,7]. Current studies have indicated that seizing sentinel lymph nodes (LN) is essential for further tumor dissemination [8], implying that susceptible immune cells are an indispensable element of metastasis progression. Immune cells exhibit substantial diversity and plasticity, and they respond to surrounding signals by earning specific functional phenotypes that can either suppress or promote metastasis. In the initial stages of tumor formation, cytotoxic immune cells, such as natural killer (NK) and CD8+ T cells, recognize and destroy highly immunogenic tumor cells [9]. However, tumor heterogeneity may present a subpopulation of cancer cells which has less immunogenic, leading to the escape of the immune attack, while the associated mechanism remains unclear. As neoplastic tissue develops into a clinically detectable tumor, these tumor cells have already evaded assault from immune cells and assimilated them [10]. Indeed, cancer cell-derived cytokines, like TGFβ and IL-10, frequently direct differentiating tumor-infiltrating immune cells toward a tumor-promoting phenotype [11]. These tumor-educated myeloid cells, particularly tumor-associated macrophages (TAMs), are able to retard anti-tumor immune responses through the synthesis of immunosuppressive cytokines, the expression of T cell coinhibitory molecules, and the consumption of amino acids that are critical for the activity of effector T cells [12]. Although clinical evidence has shown that a high level of tumor-infiltrated T cells correlates with a good prognosis in various solid cancers [13], a high level of macrophage infiltration is substantially associated with a worse prognosis [14,15,16]. However, the impact of the tumor microenvironment of reprogramming macrophages on tumor development and progression remains unclear. Hence, a better understanding of the underlying molecular mechanisms of the tumor micro-environment in macrophage differentiation and their influence on oncogenic enhancement would help identify alternative therapeutic regimens to combat peritoneal metastasis in cancer.

Tumors have a complex multi-cellular ecosystem that facilitates the malignant potential of tumor development. In this ecosystem, innate immune cells are highly abundant. Among the most abundant immune cells, macrophages are the predominant population [17,18]. As the first line of defense of the host, macrophages are specialized antigen-presenting cells (APC). They engulf and present foreign materials to T cells through major histocompatibility complex (MHC) class I and II molecules. Vigilant homeostasis should always be maintained between macrophages and T cells. Dysregulation of this order can lead to immunodeficiency causing damage to host tissues in the form of autoimmunity, which can trigger cancer development and lead to unsuccessful eradication of invading pathogens [19]. Inevitably, TAMs mainly manifest as alternatively activated M2 macrophages in the tumor mass, where they generally exhibit protumor effects by facilitating tumor survival, proliferation, angiogenesis, and dissemination [20]. Macrophages also potentiate cancer cell dissemination and tumor initiation when inflammation is a causal factor [21]. Emerging evidence suggests that tumor development, progression, and metastasis are influenced by dynamic changes in macrophage polarization. The defined subpopulations of macrophages are responsible for tumor-promoting activities [22,23,24,25]. Hence, targeting TAMs as novel immunotherapeutic strategies to stop tumor progression and metastasis has attracted increasing attention in recent years. To date, numerous potent molecular candidates for TAM-targeted therapies are under investigation.

In this review article, we aim to provide a brief synopsis of the impact and correlated functions of the tumor micro-environment on macrophage diversity polarization. We also aim to explore the associated mechanisms in metastatic progression, as well as the latest therapeutic agents for potential TAM-targeted therapies.

## 2. Macrophage Phenotypes

Macrophages are versatile immunocytes that execute a broad spectrum of functions that range from governing tissue homeostasis, defending against pathogens, and helping wound healing [26,27]. Depending on the physiological micro-environments in which they are embedded [20,28,29], macrophages have distinct presentations and even exhibit opposing phenotypes. The most popular classifications have been applied to the subtype dichotomy in immunological responses (Figure 1). These include “activated” macrophages that are associated with the responses of type I helper T (Th1) cells to pathogens. They are activated by interferon-gamma (IFNγ) and the engagement of Toll-like receptors (TLR) by exhibiting increased expression of major histocompatibility complex (MHC) class II, interleukin (IL-12), and tumor necrosis factor α (TNFα); generate reactive oxygen species (ROS) and nitric oxide (NO); and have the ability to kill pathogens [30,31]. In contrast, “alternatively activated” macrophages show a different response to IL-4 and IL-13 and are involved in Th2-type responses, including humoral immunity and wound healing [32,33]. Other macrophage populations are involved in tissue development and homeostasis, which are primarily regulated by CSF-1, and do not fall easily into these immunological categories [21]. For example, macrophages infiltrate different tissues with different statuses, such as splenic red pulp macrophages, large peritoneal macrophages, brain microglia, and Kupffer cells in the liver, and exhibit more differences in their transcriptional program than similarities [27,34]. Mantovani et al. suggested that macrophages in tumors are M2-like macrophages instead of M1-like macrophages [21]. Current studies on TAMs suggest that macrophages undergo M2-like polarization into TAMs [35]. However, in contrast to the binary M1/M2 distinction, TAMs have consisted of several particular populations that often share features of both types but are significantly more similar overall to macrophages associated with developmental processes.

## 3. Heterogeneous TAM Polarization

Clinical studies have provided convincing evidence that macrophages can promote tumorigenesis [21]. Over 80% of previous studies showed a correlation between macrophage density and poor cancer patient prognosis [36,37,38]. These outcomes were supported by recent evidence that there is a strong relationship between poor survival and elevated macrophage density in thyroid cancer, non-small-cell lung carcinoma (NSCLC), and hepatocellular carcinomas [39,40,41]. Most of the abovementioned studies consider TAMs M2-like macrophages that secrete tumor growth factors, promote angiogenesis, and inhibit T cells from exerting tumor-promoting effects [42,43]. The bias of unbalanced TAM polarization and distribution in the tumor microenvironment is a current direction in cancer research. However, the mechanisms by which tumor cells re-educate macrophages to TAMs to exert a pro-tumor effect in promoting metastatic progression are poorly understood. Current research indicates that tumor cells re-educate macrophages in the following ways (Figure 2).

### 3.1. Cytokines

Tumor cell-macrophage crosstalk is affected by phenotypic and functional alterations of both cell types. Mounting evidence suggests that secretions from tumor cells shift the transcriptional program from the M1-like phenotype to the M2-like TAMs phenotype [20,44]. For instance, tumor cell-derived C-C motif chemokine ligand 2 (CCL2) and colony stimulating factor 1 (CSF1) lead to increased infiltration of macrophages in the tumor microenvironment (TME), which, in turn, enhances angiogenesis by inducing vascular endothelial growth factor (VEGF) expression [45,46]. Furthermore, experiments using co-cultured tumor cells and macrophages showed augmented expression levels of IL10, IL12, IL6, TNF, CCL5, CCL22, and CSF1 in macrophages, which facilitated M2-like polarization [47]. Indeed, numerous studies have reported that the overexpression of CSF-1 and CCL2 is correlated with poor prognosis in numerous human cancers, including breast, ovarian, endometrial, prostate, hepatocellular, and colorectal cancer, etc. [15,48]. These studies strongly indicate that particular growth factors and chemokines play important roles in tumor macrophage biology [49]. It is noteworthy that reinforcing receptors of these cytokines in tumor cells can also promote a malignant phenotype [50,51,52]. For example, Zabuawala et al. demonstrated that genetic ablation of the Est-2 transcription factor, which is a direct effector of the CSF-1 pathway, in myeloid cells resulted in suppression of metastasis in both the transgenic polyoma middle T oncoprotein (PyMT) and orthotopic transplant breast cancer models [21,53]. These data indicate that crosstalk between TAMs and tumor cells can enhance cancer cell malignancy [54,55]. Cassetta et al. reported that an autoregulatory loop between TAMs and cancer cells is governed by tumor necrosis factor-alpha (TNFα), CCL8 and SIGLEC1, and is self-reinforced through CSF1 production [25]. Secreted factors, such as cytokines and chemokines, thus bridge the crosstalk between TAMs and tumor cells [56], playing a crucial role in promoting tumor growth, progression, metastasis, and therapeutic resistance [57,58,59,60].

### 3.2. Metabolites

The nutritional demands of tumor cells and the TME perturbations have a subtle impact on not only TAM survival but also cancer progression and anti-tumor immune responses [46,61]. Different metabolic patterns indicate the diverse functions of macrophages. M1-like macrophages remove pathogens by generating inflammatory factors, such as IL12 and nitric oxide (NO), which require high glycolytic metabolism [46,62]. Conversely, protumor (M2-like) macrophages are generally considered to exploit oxidative metabolism for bioenergetic purposes, which has been determined by their ability to support tissue repair [46,63]. This oversimplified view does not properly reflect macrophage metabolic heterogeneity but proposes that tumor-suppressing and tumor-promoting functions rely on different metabolic pathways [64,65]. For instance, M1 macrophages utilize arginine to generate nitric oxide (NO) to kill pathogenic substances by expressing the enzyme nitric oxide synthase [66]. On the other hand, M2 macrophages express arginase enzymes to hydrolyze arginine to ornithine and urea and facilitate tissue repair [66]. Similar considerations apply to TAMs, mainly M2-like TAMs, which release trophic factors that promote oxidative phosphorylation (OXPHOS) and fatty acid oxidation (FAO) instead of glycolysis to meet the needs of biosynthesis [46]. Cancer cells can harness metabolic byproducts to hijack the functions of tumor-infiltrating immune cells to their benefit. Wu et al. recently showed that cancer cells secrete succinate into the extracellular milieu, which mediates TAM polarization and promotes cancer metastasis [22]. Moreover, Zhou et al. reported that all-trans retinoic acid (ATRA) inhibits osteosarcoma metastasis by inhibiting the M2 polarization of TAMs [67]. These studies indicate that tumor cells can modulate metabolites to activate the immunosuppressive TME. Moreover, macrophages can also be polarized to TAMs by adapting to the TME. For instance, Anders Etzerodt et al. reported that CD163+ TAMs infiltration occurs in melanomas and is associated with the resistance of PD-1/PD-L1 therapy [68]. Studies indicated that nutrient deprivation in the TME impairs the functions of T cells or NK cells [69], and enforces macrophage polarization into TAMs through reprogramming their metabolic patterns [70]. Recent studies have shown that TAMs express elevated levels of the scavenger receptor CD36, accumulate lipid droplets, and utilize fatty acid oxidation (FAO) for energy generation to support tumor growth [71]. Goossens et al. reported that cancer cells scavenge membrane cholesterol from TAMs, leading to their reprogramming toward immune suppressive and tumor-promoting characteristics [72]. In fact, targeting cholesterol efflux in TAMs is able to inhibit this reprogramming and reduces tumor progression in ovarian cancer [72]. This suggests a crosstalk-mediated relationship related to energy supply between TAMs and tumor cells.

### 3.3. Exosomes

Exosomes are another type of TAMs inducer. Emerging evidence has suggested that exosomes act as bridges connecting different components of the TME [73,74,75,76]. Exosomes are extracellular vehicles (EVs) with 30–150 nm diameters. They mainly consist of proteins and RNAs and sometimes include glycoconjugates, lipids, and DNAs [77]. Exosomal proteins include integral exosomal membrane proteins, lipid-anchored outer membrane proteins, peripheral surface proteins, lipid-anchored inner membrane proteins, inner peripheral membrane proteins, exosomal enzymes, and soluble proteins [78]. Exosomal RNAs include mRNA and noncoding RNAs, i.e., microRNAs and long noncoding RNAs (lncRNA). Previous studies have shown that in a variety of cancers, the exosomes of cancer cells can be secreted into the TME to alter the function of neighboring cells, thereby creating an adaptive pre-metastatic niche conducive to tumor development [78,79,80]. For example, Chen et al. demonstrated that in hypoxia-induced epithelial ovarian cancer (EOC), microRNA-940 (miR-940) is secreted from exosomes and promotes M2-like TAM polarization, which in turn, accelerates EOC proliferation and migration [81]. Moreover, exosomal miR-222-3p secreted from ovarian cancer cells is an effective regulator of M2-like TAM polarization that supports cancer development [82]. In addition, miR-146a was reported to be enriched in exosomes and can enhance M2-like TAM polarization and suppress T cell function in HCC [83]. Another source of EVs secretion is tumor-infiltrated Mesenchymal Stem Cells (MSC) derived from bone marrow for promoting regeneration, immune adaptation, and modulation of the TME [84]. In fact, a recent study showed that tumor-infiltrated MSCs secrete EVs for facilitating M2-like TAM polarization and promoting breast and gastric cancer (GC) progression [85].

On the other hand, M2-polarized TAMs can also accelerate tumor aggression and progression by their secreted exosomes. Zheng et al. has recently reported that M2-like TAMs enhance GC cell migration and invasion by secreted exosomal Apolipoprotein E in activating the PI3K-AKT signaling in GC cells [76]. Within the TME, tumor cells communicate with macrophages through exosomes to maintain a vicious cycle to facilitate tumor progression [86,87]. This evidence shows that tumor cells communicate with TAMs through exosomes, forming a positive loop to escalate tumor progression.

## 4. TAMs are Involved in Tumor Progression

Previous studies using experimental models have shown that depletion of macrophages can inhibit tumor progression and metastasis. On the other hand, it is also known that macrophages inherent anti-tumor potential through antigen presentation and secretion of anti-tumor factors, active T cells, and NK cells [9,88]. Hence, it is necessary to clarify their functions by determining the mechanisms of macrophage polarization and the functions of macrophages during tumor development and progression (Figure 3).

### 4.1. Macrophages in Cancer Initiation

Chronic inflammation or oncogene activation are often involved in the production of cytokines and chemokines that in turn, engage the innate immune system, especially macrophages [57]. Inflammatory macrophages play an important role in cancer initiation by creating a mutagenic micro-environment through producing proinflammatory mediators, such as IL-6, TNFα, and interferon-gamma (IFNγ); as well as growth factors, including epidermal growth factor (EGF) and metabolites [21,52]. Thus, chronic inflammation is involved in the initiation of several solid tumors, such as cancers of the colon or stomach [89]. However, the dynamic polarization of macrophages in tumor development and progression has not been well-studied [17]. Studies have indicated that macrophages are mainly tumoricidal at the initial phases of tumor formation (or early stages) due to the response to T helper 1 (Th1)-like inflammatory signals [38,52]. However, non-malignant cells, such as macrophages, evolve along with the tumor, and provide essential support for tumor development and progression in the microenvironment [90,91]. During tumor development from benign to invasive stages, the microenvironment is enriched with cytokines and growth factors, leading to a bias away from converting the T helper 1 (Th1)-like inflammatory response to a Th2-type immune environment (Figure 3). Once tumors are formed, macrophages are educated to M2-like TAMs to promote tumor growth and exert immunosuppressive effects. This Th2 environment is featured by transforming growth factor-β1 (TGF-β1) and Arginase 1 and increased numbers of CD4+ T cells [17]. However, it is unclear when this transition occurs and whether macrophages have an antitumoral ability to eradicate aberrant mlignant cells before the formation of tumor.

### 4.2. TAMs Facilitate Tumor Metastasis and Intra-tumoral Heterogeneity 

Highly invasive tumor cells derived from the primary tumor intravasate into the blood circulation or lymphatic vessels, disseminating to distant sites to cause micro-metastases. Mounting evidence has suggested TAMs are proactively involved in this process by promoting metastatic cancer cells spread via a paracrine loop existing between TAMs and tumor cells [20,92,93,94]. Previous studies indicated that macrophage infiltrated tumor mass enhances tumor invasion by secreting EGF ligands [95]. In addition, TAMs can release cathepsins and matrix-remodeling enzymes like MMP9 to assist tumor cells in modulating ECM at the invasion site [96]. On the other hand, metastatic cancer cells recruit TIE2 receptor-expressing monocytes by angiopoietin-2 that, in turn, sensitize hypoxia and produce angiogenic factors to enhance angiogenesis [97]. Recently, Huang et al. have found spatial heterogeneity of TAMs in tumor mass where the CD68+IRF8+TAMs (M1-like) are wrapped in inner regions of tumor mass, and the CD68+CD163+CD206+ TAMs (M2-like) are enriched at tumor peripheral regions [29]. This distribution pattern of TAMs indicates that M2-like TAMs functions to exert immune suppression in the TME and assist tumor invasion, while M1-like TAMs may be involved in necrosis in the central core of tumors. Clinically, the above structures can be used to calculate a prognostic clinical score to predict the chance of metastasis in breast cancer regardless of tumor stage or subtype [98]. In addition, previous evidence has shown that TAMs play an essential role in the survival and proliferation of metastatic cancer cells detached from the primary tumor in the peritoneal environment and tumor spheroids at early stages of peritoneal metastasis in mouse orthotopic ovarian cancer models [99]. However, the significance and mechanism of these immune cells, such as TAMs, in assembling tumor spheroids in the malignant ascites is still unclear.

Numerous neoplastic intrinsic and extrinsic mechanisms are involved in tumor progression (Figure 3); however, the mechanisms driving the heterogeneity of neoplastic cells in solid tumors remain unclear. Increased mutational rates of neoplastic cells in stressed environments are implicated but cannot be applicable for all aspects of tumor heterogeneity [100,101]. A previous study by Gast1 et al. showed that fusion of tumor cells with TAMs causes tumor heterogeneity and promotes metastatic capacity [102]. More importantly, Charles et al. confirmed that hybrid cells (macrophage-tumor cells) exist in human tumor biopsies and the peripheral circulatory system [102]. These divergent views cause controversy regarding whether cell fusion provides a selective advantage to evolving tumors.

## 5. Immunotherapy and TAM-targeted Therapy

The current treatment strategies for advanced malignancies are far from effectively preventing cancer-related deaths. Immunotherapies have recently provided hope as efficacious and novel therapeutic regimens for advanced cancers. However, the current clinical data indicate that therapeutic responses are suboptimal and vary substantially among individuals [49]. Among the possible reasons for the failure of immunotherapies, the existence of TAMs in the tumor microenvironment (TME) is the crucial factor because of their significant correlation with poor prognosis and therapy resistance [103]. Thus, an in-depth investigation of the complicated roles of TAMs in the TME and their impact on metastatic progression is urgently needed to improve the efficacy of immunotherapy. TAM-targeted therapies have become emerging strategies for treating cancer and chemoresistance.

### 5.1. Inhibition of TAMs Recruitment or Reprogramming of TAMs

Macrophages cannot destroy tumor cells directly when simply activated by interferon-γ (IFN-γ) but need to recruit activated CD8+ cytotoxic T lymphocytes and NK cells by presenting antigens and secreting cytokines [104,105]. T cells activate macrophages through the CD40-CD40L interaction to enhance the expression of major histocompatibility complex class II (MHC II), inducible nitric oxide (iNOS), and TNF [106]. However, TAMs inducers are usually dominant in the TME, and they activate macrophages and induce their polarization to the M2-like phenotype, impeding effector T cells from invading inside the tumor [17,107]. As a result, the best solution is to hinder the recruitment of macrophages by the tumor mass, which inhibits TAM polarization (Figure 4, Table 1). Evidence has proven that TAMs accumulate in the tumor mass through the CCL2–CCR2 axis [108]. Indeed, targeting the CCL2-CCR2 axis has been proven effective in reducing tumor growth and metastasis in mouse models [109]. BMS-813160 is a CCR2/CCR5 dual antagonist that can inhibit regulatory T cell and myeloid-derived suppressor cell infiltration. Its effect has been studied in combination treatment in NSCLC, hepatocellular carcinoma, and pancreatic ductal adenocarcinoma (NCT04123379, NCT03496662). Moreover, the CXCL12/CXCR4 signaling cascade promotes cancer progression and metastasis and regulates TAM recruitment [110,111]. The effect of the CXCR4 antagonist BL-8040 combined with pembrolizumab in metastatic pancreatic cancer is being evaluated (NCT02907099). In addition, substantial evidence indicates that the CSF-1/CSF-1R axis is an attractive target for reducing the number of TAMs in tumors [35]. Previous research has indicated that the number of CSF1R+ macrophages correlates with poor survival in various tumor types [112,113,114]. In 2019, pexidartinib, a CSF1R inhibitor, was approved by the U.S. FDA for the treatment of giant-cell tumors (GC-TS). Moreover, the CSF1R inhibitor SNDX-6352 combined with durvalumab is under evaluation in a phase 2 trial (NCT04301778). Recently, TPX-0022, a MET/CSF1/SRC inhibitor, was tested in phase 1 trials in patients with advanced solid tumors harboring MET genetic alterations (NCT03993873). However, adverse reactions restrict the clinical application of CSF1R inhibitors. For instance, the side effects of pexidartinib include increased aspartate aminotransferase levels, fatigue, nausea, and potential liver toxicity [115]. In addition, direct depletion of TAMs by zoledronate or clodronate in mouse models can delay tumor progression [116,117], while several clinical trials have demonstrated inconsistent results, implying that this strategy needs to be further optimized [52]. Depletion of immune cells might cause severe bacterial infections, especially in the context of cancer [118]. Therefore, the complete deletion of immune cells is not desirable for clinical cancer treatment.

Due to the plasticity of macrophages, retrofitting TAMs has become another pharmaceutical option. The current research direction is to reprogram M2-like TAMs to the M1-like phenotypes (Table 2). Based on this concept, Hu et al. identified approximately 30 compounds that can switch M1-like phenotypes toward the M2-like TAMs and that another ~20 compounds can switch M2-like TAMs to the M1 phenotype through high-throughput screening [119]. Furthermore, they showed that thiostrepton could reprogram TAMs toward an M1-like state in mice and has the anti-tumor ability [119]. In addition to pharmacological approach, oligonucleotide delivery is also a promising and popular method to manipulate TAMs. Klichinsky et al. found that a chimeric adenoviral vector overcomes the inherent resistance of primary human macrophages to genetic manipulation and causes macrophages to adopt a sustained proinflammatory (M1-like) phenotype [120]. CAR macrophages (CAR-M) demonstrate antigen-specific phagocytosis and tumor clearance in vitro [120]. Characterization of CAR-M activity has shown that CAR-Ms express proinflammatory cytokines and chemokines, convert bystander M2-like TAMs to M1-like phenotype, promote antigen presentation machinery, recruit and present antigens to T cells, and resist the effects of immunosuppressive cytokines. In humanized mouse models, CAR-Ms have further been shown to induce a pro-inflammatory tumor micro-environment and boost anti-tumor T cell activity [120]. In addition, nanoparticle encapsulation is another innovative strategy for macrophage reprogramming. C Wyatt Shields 4th et al. reported an engineered particle known as “backpack” that could remarkably adhere to the macrophage surface and control cellular phenotypes in vivo [121]. Similarly, Zhang et al. described a nanocarrier-based delivery system for loading interferon regulatory factor 5 and IKKβ to TAMs to reverse M2-like TAMs into M1-like phenotypes without causing in vivo systemic toxicity [122]. In addition, Wei et al. treated macrophages with mannose (Man)-modified macrophage-derived MPs, which are carriers for the targeted delivery of metformin (called Met@Man-MPs) to M2-like TAMs [123]. They showed that Met@Man-MPs efficiently reset TAMs toward the M1 phenotype to inhibit tumor growth. Met@Man-MPs significantly improved the tumor immunosuppressive microenvironment and enhanced CD8+ T cell infiltration into the tumor interior by restoring macrophage-induced recruitment of CD8+ T cells and Man-MP-induced tumor ECM degradation because macrophages express matrix metalloproteinases (MMP) [123]. In addition, the utilization of normal MSCs to modulate TAM polarization is another promising therapeutic strategy. Noemí Eiró et. al found that the conditional medium of the normal human uterine cervix (hUCESC) or MSCs could reprogram TAM polarization into M1-like population [124]. Interestingly, the conditional medium contains secreted EVs from MSCs exerting anti-tumor function [125]. Using the EVs isolated from normal MSCs could increase M1-like macrophage population, reducing the risk of tumor progression and aggression in the TME. Therefore, reprogramming of TAMs in the TME has become a hot topic in research on tumor immunotherapy.

### 5.2. Phagocytosis Checkpoints

Homeostasis maintenance eliminates non-self cells, and normal cells can avoid self-elimination by phagocytes by expressing anti-phagocytic molecules [127] to pass “phagocytosis checkpoints” [128]. However, previous studies have shown that tumor cells prefer phagocytosis checkpoints to evade immune surveillance [129]. Signal regulatory protein alpha (SIRPα) is an ITIM-bearing inhibitory receptor displayed on myeloid cells, including macrophages. SIRPα recognizes CD47, which acts as a “don’t eat me” signal. It has been found to be overexpressed in tumor cells, and its expression correlates with poor patient survival [35]. Macrophage-mediated phagocytosis of tumor cells is restored after treatment with CD47 antibodies, and this macrophage-mediated phagocytosis process is further enhanced in the presence of chemotherapeutic drugs, suggesting that lower CD47 expression are more likely to benefit from adjuvant TACE treatment [35,130]. Moreover, the CD24-Siglec-10 axis was recently identified as a new phagocytic checkpoint. Genetic depletion of either CD24 or Siglec-10 or blockade of the CD24–Siglec-10 interaction by monoclonal antibodies can markedly augment the phagocytosis of human breast and ovarian cancer cells [131]. However, it is still unknown whether phagocytosis checkpoint activation in TAMs alone is sufficient to control tumor growth or whether the involvement of T cells is required. Thus, further investigation is needed.

### 5.3. cGAS-STING in TAMs

The cGAS–STING signaling axis recognizes pathogenic DNA and induces type I interferon production. These events promote the cross-priming of macrophages and initiate a tumor-specific CD8+ T cell response [132,133]. Miao et al. identified STING as a prognostic factor for gastric cancer. They demonstrated that inhibition of STING or 2′3′-c-GAMP can promote TAM polarization to a pro-inflammatory subtype and induce apoptosis of gastric cancer cells mechanistically through the IL6R-JAK-IL24 pathway [134]. Furthermore, Cheng et al. utilized liposomal nanoparticle-delivered cGAMP (cGAMP-NP) to boost immune responses in triple-negative breast cancer and melanoma [126]. A mechanistic study indicated that cGAMP-NPs promote TAM polarization to an M1-like phenotype and increase T cell number and infiltration in vivo [126]. Likewise, Lv et al. found that manganese is dispensable for the activation of the cGAS-STING cascade to achieve a tumor-suppressive function [135] by boosting the anti-tumor response of a wide variety of immune cells, such as NK cells, DC cells, macrophages, and T cells [135]. These findings collectively suggest that manipulating the anti-tumor effect of cGAS-STING is currently a hot spot in research. However, chromosome instability enhances tumor metastasis through the rupture spills genomic DNA from micronuclei that in turn, activates cGAS-STING pathway [136]. However, the activation of this pathway caused by chromosomal instability could not eliminate the tumor and even makes it more malignant [136]. Therefore, how to utilize the anti-tumor effect of cGAS-STING without promoting metastasis is worthy of further research. Another problem worthy of study is whether regulating cGAS-STING in TAMs can prevent chronic inflammation and facilitate anti-tumor function.

## 6. Conclusions, Perspectives, and Limitations

There is a complicated interaction between tumor cells and TAMs in the tumor mass. Macrophages are responsible for engulfing and killing tumor cells. However, many TAMs that infiltrate the tumor micro-environment do not kill tumor cells but support tumor development and metastatic progression. Particularly, tumor cells secrete many chemokines, such as CCL2, to recruit macrophages into tumor tissues. After macrophages are recruited to tumor tissue, tumor cells secrete various cytokines, metabolites, and exosomes to alter the functions and polarization of TAMs. Eventually, TAMs differentiate into M2 cells, which sustain tumor growth by facilitating the proliferation, invasion, and metastasis of malignant cells; consume effector T cells; and promote tumor angiogenesis. Importantly, these events are intimately related to the inflammatory TME. Regulating the inflammatory TME is the key to improving the immunosuppressive TME. The acute inflammatory response and the transformation of the chronic inflammatory response at the beginning of the tumor are worthy of study. There are also numerous strategies for limiting the supporting effect of TAMs on tumor growth. The current primary strategy is to inhibit the recruitment of macrophages, which reduces the accumulation of TAMs in tumor tissues and targets TAM receptors, or to block key cytokines secreted by tumor cells to reprogram TAMs to M1 cells, which have anti-tumor activity. However, TAMs inhibit toxic T cell function and limit the response to current immunotherapies.

In the future, the multi-target design strategy on TAMs should be considered to reprogram the TME to improve the response of immunotherapy. Understanding the dynamics of macrophage polarization and how these dynamics affect the functional phenotype of tumor-infiltrating TAMs is vital for the development and improvement of therapies. A thorough exploration of TAM polarization mechanisms will provide a better understanding of the pathophysiology of tumor development and progression and potentially present new opportunities for therapy and monitoring of patients with malignant tumors. For example, multisite targeted combined drug therapy or nanoparticle gene therapy may have a better therapeutic effect. In addition, current clinical trials on TAM-targeted treatments in animals are still lacking, and further improvement is urgently needed. Because there are a large number of infiltrating TAMs inside the tumor tissues, targeting TAMs is a new direction for the treatment of solid tumors. Nevertheless, TAMs need to be retrofitted and exhibit anti-tumor properties in the future.

## Figures and Tables

**Figure 1 ijms-22-06560-f001:**
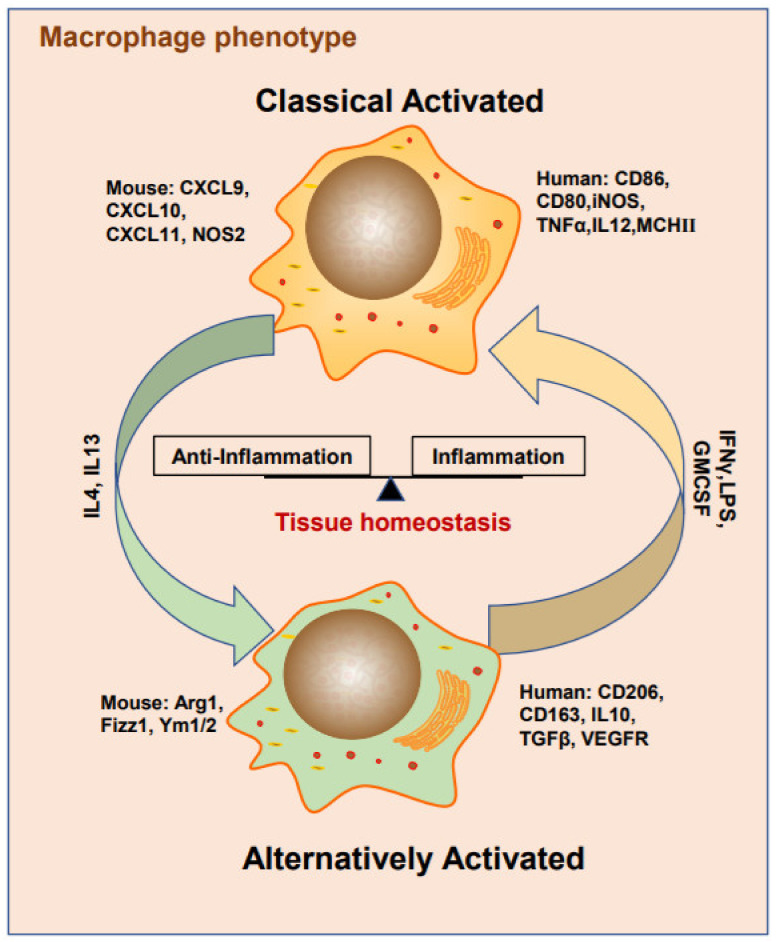
Macrophage M1/M2 polarization status maintains organizational stability. IFN-γ, LPS, GMCSF are the key stimulators of classically activated macrophages (recognized as M1). During the acute inflammation phase, M1 macrophage induces inflammatory responses by expressing cell surface markers to attract immune cells and releasing inflammatory factors. On the other hand, IL4 and IL13 are inducers of alternatively activated macrophages (recognized as M2), which switch the inflammatory response to anti-inflammatory to carry out tissue remodeling function.

**Figure 2 ijms-22-06560-f002:**
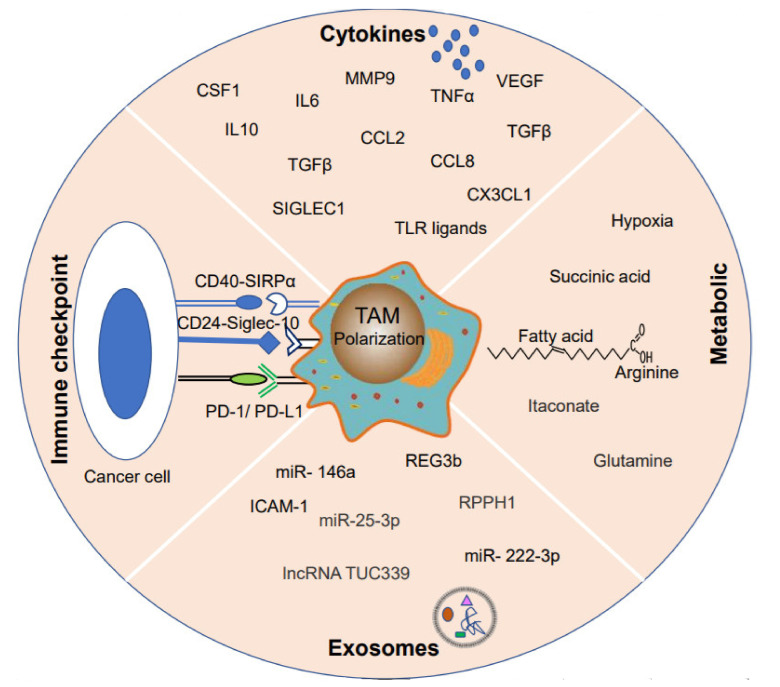
The crosstalk of tumor cells and TAMs. Tumor cells secrete different cytokines or exosomes to polarize macrophages into TAMs directly and hijacking macrophages by metabolites or limiting oxygen concentration indirectly. On the other hand, the polarized TAMs express PD-L1, SIRPα, or Siglec-10 to inhibit the anti-tumor function.

**Figure 3 ijms-22-06560-f003:**
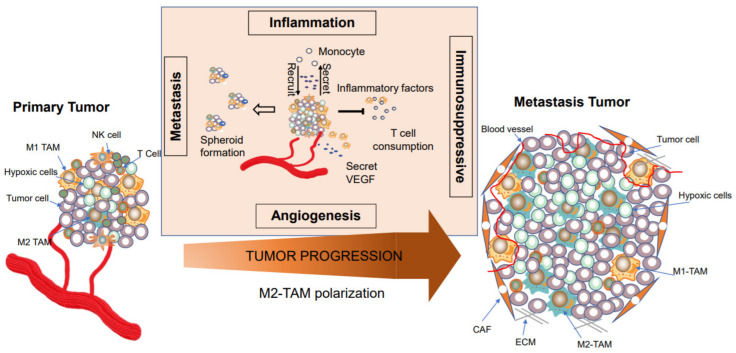
Co-evolution of cancer cells and TAMs in tumor progression. The early stage of the primary tumor exhibits more M1-like TAMs, T cells, and NK cells. Along with disease progression, tumor cells promote M2-TAM polarization and utilize M2-like TAMs to inhibit effector T cell or NK cell infiltration. Moreover, TAMs promote tumor spheroid formation and facilitates peritoneal metastasis. The polarized M2-like TAMs create the immunosuppressive micro-environment for facilitating metastatic tumor progression by secreting cytokines, such as VEGF, to induce angiogenesis, releasing inflammatory factors to recruit circulating monocytes to tumor mass, utilizing amino acids like arginine to inhibit T cells function, and promoting tumor spheroid formation to facilitate tumor metastasis.

**Figure 4 ijms-22-06560-f004:**
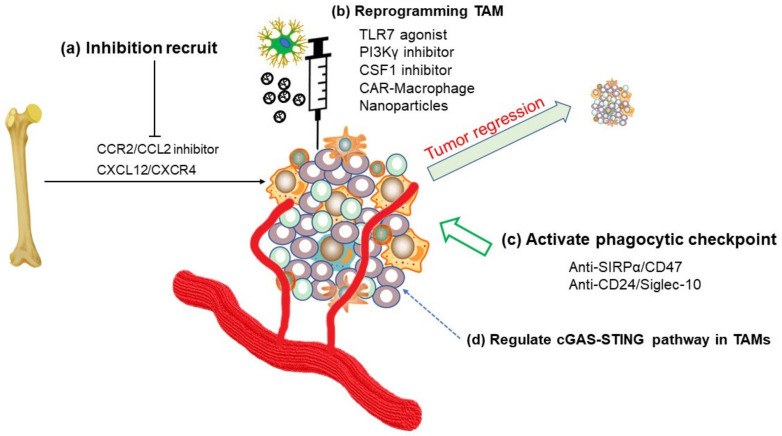
TAM-targeted therapy in tumors. Targeting TAM strategies consists of (**a**) the inhibition of the recruitment of macrophages to tumor mass; (**b**) reprogramming TAMs into M1-like macrophages to strengthen antitumor ability (**c**) the activation of “phagocytic checkpoint” to restore macrophage phagocytic function; and (**d**) targeting cGAS-STING pathway in TAMs is a potential anti-tumor approach.

**Table 1 ijms-22-06560-t001:** TAM-targeted therapy in clinical trials.

TAM Targeted Strategies	Compound	Targets	Therapy	Tumor Type	Phase	References
Inhibit the recruitment	BMS-813160	CCR2/5-inhibitor	Combination	Non-small Cell Lung Cancer or Hepatocellular Carcinoma, Pancreatic Ductal Adenocarcinoma	1,2	NCT04123379, NCT03496662
	Carlumab	anti-CCL2 antibodies	Single agent	Prostate Cancer	2	NCT00992186
	Plerixafor	CXCR4/CXCL12 inhibitor	Combination	Metastatic Pancreatic Cancer	2	NCT04177810
	BL-8040	CXCR4 antagonist	Combination	Metastatic Pancreatic Adenocarcinoma	2	NCT02907099
	DCC-3014	CSF-1R inhibitor	Single agent	Advanced Malignant Neoplasm	1, 2	NCT03069469
	SNDX-6352	CSF-1R inhibitor	Combination	Solid Tumor, Metastatic Tumor, Unresectable Intrahepatic Cholangiocarcinoma	1, 2	NCT03238027, NCT04301778
	TPX-0022	MET/CSF1/SRC inhibitor	Single agent	Advanced Solid Tumor Metastatic Solid Tumors MET Gene Alterations	1	NCT03993873
	LY3022855	CSF-1R inhibitor	Combination	Melanoma	1, 2	NCT03101254
	IMC-CS4	CSF-1 R mAb	Combination	Pancreatic Cancer	1	NCT03153410
	Cabiralizumab	CSF-1 R mAb	Combination	Advanced Melanoma, Non-small Cell Lung Cancer, Renal Cell Carcinoma	1	NCT03502330
	Regorafenib	CSF-1R	Combination or Single agent	Hepatocellular Carcinoma	1, 2	NCT04170556
Active phagocytic checkpoint	STI-6643	Anti-CD47 mAb	Single agent	Solid Tumor	1	NCT04900519
	Hu5F9-G4	Anti-CD47 mAb	Combination	Hematological Malignancies	1	NCT03248479
	TTI-621	Anti SIRPαFc	Combination or Single agent	Hematologic Malignancies or Solid Tumor	1	NCT02663518

**Table 2 ijms-22-06560-t002:** TAM-targeted therapy in preclinical studies.

TAM Targeted Strategies	Method	Therapy	References
Reprogramming M2-like into M1-like phenotypes	Thiostrepton	M1-activating compound could reprogram M2-like into M1-like phenotypes	[119]
	CAR-M	Chimeric adenoviral vector transfer macrophage to proinflammation status (M1-like)	[120]
	Nanoparticle	Nanocarrier delivery interferon regulatory factor 5 and IKKβ to polarize macrophage to M1-like phenotypes	[122]
	EVs	Isolate normal MSCs derived Evs or conditional medium to switch M2-like to M1-like phnotypes	[124,125]
Regulation of cGAS-STING pathway in TAMs	Nanoparticle	Liposomal nanoparticle-delivered cGAMP to TAMs to promote M1-like polarization	[126]

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
