# Peer review of "The Impact of the Tumor Microenvironment on Macrophage Polarization in Cancer Metastatic Progression"

_ijms, 2021, doi:10.3390/ijms22126560_

Round 1

Reviewer 1 Report

The concept of TAMs are well summarized in this article, however, additional description should be added in the text.

Comment 1; Macrophages in actual cancer patients often exhibit both characteristics with mixed M1/M2-related cytokine expression, and they fall into more complex subgroups rather than two distinct patterns. Although the M1/M2 paradigm is presently accepted and applied as a conceptual framework, it is more appropriate to use terms such as “M1-like” and “M2-like”, rather than “M1” and “M2”, based on the relative expression levels of each marker in vivo. It is important to note that the M1/M2 paradigm is now considered an oversimplified concept, and that phenotypes of macrophages do not necessarily reflect their pro- and anti-inflammatory activities.

Comment 2; The paragraph of 4(lane 241-246) is not enough. Could you add the contents of CD163, which is a well-known marker for M2-like TAMs?

Author Response

Comment 1; Macrophages in actual cancer patients often exhibit both characteristics with mixed M1/M2-related cytokine expression, and they fall into more complex subgroups rather than two distinct patterns. Although the M1/M2 paradigm is presently accepted and applied as a conceptual framework, it is more appropriate to use terms such as “M1-like” and “M2-like”, rather than “M1” and “M2”, based on the relative expression levels of each marker in vivo. It is important to note that the M1/M2 paradigm is now considered an oversimplified concept, and that phenotypes of macrophages do not necessarily reflect their pro- and anti-inflammatory activities.

Response: Thanks for your valuable suggestion. We agreed that M1/M2 paradigm is an oversimplified concept, and TAM in tumor patients does not fall into this binary opposition classification. We have revised M1, M2 term as M1-like or M2-like, and the relevant statement in the ‘Macrophage phenotype’ part in the revised version.

Comment 2; The paragraph of 4(lane 241-246) is not enough. Could you add the contents of CD163, which is a well-known marker for M2-like TAMs?

Response: Thanks for the excellent suggestion. We have added an example that illustrated CD163+TAM contributes to the resistance of immunotherapy to melanoma.

Reviewer 2 Report

Wang et al´s review addresses a very interesting topic of great potential interest to journal readers, such as the role of macrophages in tumor progression, their activation mechanisms in the tumor context, and proposals for strategies that face their consideration as therapeutic targets. However, the review article does not make a significant contribution to this field of knowledge about macrophages in the progression of human carcinomas, nor original approaches that guide future research. Furthermore, although the title includes the term “… in metastatic peritoneal cancer”, this concept is barely discussed throughout the review. On the other hand, I found no prior references to the authors´ published work on the subject. The article is generally well written. Specific suggestions should be taken into account by the authors:

-Although the title includes the term "... in metastatic peritoneal cancer", this concept is hardly addressed throughout the review. Greater emphasis should be placed on the role of tumor-associated macrophages in this particular type of tumor progression.

-In my view, the work should be more conveniently structured with related sections and subsections, avoiding repetitive information.

-I believe that Figure 2 should refer to the different pro-tumor mechanisms associated with the groups of represented intercellular signal mediators.

-Figure 3 does not seems to explain the mechanisms by which tumor-associated promote tumor progression. This figure and figure 4 could be merged in a single one.

-I think it is convenient to prepare a table that contains the different strategies that consider TAMs as a therapeutic target for cancer, reflecting the different mechanisms and their corresponding bibliographical references.

-I recommend authors to highlight the potential interest of mesenchymal stem cells and the products derived from their secretome, such as exosomes, as immunoregulators of macrophages (for example: Eiró et al., Oncotarget 2014).

Author Response

  1. Although the title includes the term "... in metastatic peritoneal cancer", this concept is hardly addressed throughout the review. Greater emphasis should be placed on the role of tumor-associated macrophages in this particular type of tumor progression.

Response: Thanks for the suggestion. We have modified the title accordingly.

  1. In my view, the work should be more conveniently structured with related sections and subsections, avoiding repetitive information.

Response: We have amended the structure of this review.

  1. I believe that Figure 2 should refer to the different pro-tumor mechanisms associated with the groups of represented intercellular signal mediators.

Response: We have revised Figure 2 and figure legend to describe the crosstalk of cancer cells and TAM.

  1. Figure 3 does not seems to explain the mechanisms by which tumor-associated promote tumor progression. This figure and figure 4 could be merged in a single one.

Response: We have combined Fig.3 and Fig.4 into one figure to show TAM promoting tumor progression through multiple functions.

  1. I think it is convenient to prepare a table that contains the different strategies that consider TAMs as a therapeutic target for cancer, reflecting the different mechanisms and their corresponding bibliographical references.

Response: We have added two tables summarized the TAM targeting therapy in the clinical trials and in preclinical studies.

  1. I recommend authors to highlight the potential interest of mesenchymal stem cells and the products derived from their secretome, such as exosomes, as immunoregulators of macrophages (for example: Eiró et al., Oncotarget 2014).

Response: We have added the MSCs infiltrate into the tumor for facilitating TAM polarization in 3.b and 8. Relevant literature has been cited to support the point of view (Eiró et al., Oncotarget 2014).

Round 2

Reviewer 2 Report

The manuscript has been improved to the extent necessary to justify its publication.